

# Neurological phenotypes and treatment outcomes in Eagle syndrome: systematic review and meta-analysis

Melika Hassani[1], Elisabeth Waldemar Grønlund[1], Simon Sander Albrechtsen[1] and Daniel Kondziella[1,2]

[1] Department of Neurology, Rigshospitalet, Copenhagen University Hospital, Copenhagen, Denmark
[2] Department of Clinical Medicine, University of Copenhagen, Copenhagen, Denmark

Corresponding author
Daniel Kondziella,
daniel_kondziella@yahoo.com

## ABSTRACT

**Background:** Eagle syndrome is caused by an elongated styloid process affecting carotid arteries and cranial nerves. Pain, dysphagia, tinnitus, paresthesia (classic subtype), and neurovascular events (vascular subtype) may be triggered by head movements or arise spontaneously. However, Eagle syndrome remains underappreciated in the neurological community. We aimed to determine the most common neurological and non-neurological clinical presentations in patients with Eagle syndrome and to assess the clinical outcome post-surgical resection in comparison to non-surgical therapies.

**Methodology:** We conducted a systematic review of patient-level data on adults with Eagle syndrome, following PRISMA guidelines. We extracted data on demographics, presenting symptoms, neurological deficits, radiological findings, and treatments, including outcomes and complications, from studies in multiple indexing databases published between 2000 and 2023. The study protocol is registered with PROSPERO.

**Results:** In total, 285 studies met inclusion criteria, including 497 patients with Eagle syndrome (mean age 47.3 years; 49.8% female). Classical Eagle (370 patients, 74.5%) was more frequent than vascular Eagle syndrome (117 patients, 23.5%, $p < 0.0001$). Six patients (1.2%) presented with both variants and the subvariant for four patients (0.8%) was unknown. There was a male preponderance (70.1% male) in the vascular subtype. A history of tonsillectomy was more frequent in classic (48/153 cases) than in vascular (2/33 cases) Eagle syndrome (Odds Ratio 5.2, 95% CI [1.2–22.4]; $p = 0.028$). By contrast, cervical movements as trigger factors were more prevalent in vascular (12/33 cases) than in classic (7/153 cases) Eagle syndrome (Odds Ratio 7.95, 95% CI [2.9–21.7]; $p = 0.0001$). Headache and Horner syndrome were more frequent in vascular Eagle syndrome and dysphagia and neck pain more prominent in classic Eagle syndrome (all $p < 0.01$). Surgically treated patients achieved overall better outcomes than medically treated ones: Eighty-one (65.9%) of 123 medically treated patients experienced improvement or complete resolution, while the same applied to 313 (97.8%) of 320 surgical patients (Odds Ratio 1.49, 95% CI [1.1–2.0]; $p = 0.016$).

**Conclusions:** Eagle syndrome is underdiagnosed with potentially serious neurovascular complications, including ischemic stroke. Surgical treatment achieves better outcomes than conservative management. Although traditionally the domain of otorhinolaryngologist, neurologist should include this syndrome in differential

diagnostic considerations because of the varied neurological presentations that are amenable to effective treatment.

# INTRODUCTION

Eagle syndrome is a neurovascular compression syndrome that is caused by an elongated styloid process and/or calcification of the stylohyoid ligament compressing the adjacent vascular and neuronal structures. The styloid process is a thin, sharp, and columnar-shaped osseous projection arising from the petrous portion of the temporal bone, situated just anterior to the stylomastoid foramen (*Sham et al., 2020*). Embryologically, it is derived from the Reichert's cartilage of the second branchial arch, giving rise to the following four segments: the styloid process itself, the stylohyoid ligament, the lesser cornu of the hyoid bone and the hyoid bone (*Fini et al., 2000*).

The association between pain and an elongated styloid process was described by Watt Eagle in 1937 (even though others had recognized this association before him) (*Eagle, 1937*). A normal styloid process in adult Europeans and Americans ranges from 20 to 30 mm in length, and a process greater than this is considered anomalous (*Kawasaki, Hatashima & Matsuda, 2012*; *More & Asrani, 2011*). An elongated styloid process has been said to occur in 4% of the general population, while only 4% of these individuals may present with symptoms that are attributable to the elongation (*Farina et al., 2021*). However, these figures, originally presented by Dr. Eagle, have been challenged since then, and it should be noted that estimating the overall prevalence of an elongated styloid process is nearly impossible due to the significant variation in diagnostic criteria across publications. The etiology of Eagle syndrome remains poorly understood, and multiple theories have been proposed. Surgical trauma, or a chronic irritation of the stylomandibular ligament, may lead to reactive ossifying hyperplasia of the styloid process but other possibilities are osseous metaplasia of the Reichert cartilage residues, persistence of mesenchymal element capable of producing bone tissue in adults, and ossification of the stylohyoid ligament related to endocrine disorders (*Sadaksharam & Singh, 2012*; *Swain et al., 2017*).

Eagle syndrome is differentiated between two phenotypes: one related to compression of cranial nerves and soft tissue, which is more common and sometimes associated with a prior tonsillectomy (classic Eagle syndrome), and another related to compression of vascular structures, *i.e.*, the carotid artery and the jugular vein (vascular Eagle syndrome) (*Wong et al., 2011*). Symptoms related to the former subtype include tinnitus, otalgia, pharyngeal pain, and pharyngeal foreign body sensation, while the latter subtype may have a clinical presentation characterized by hemiparesis, speech disturbances and syncope (*Farina et al., 2021*). This diverse range of symptoms, along with the numerous potential

differential diagnoses that follow, can make interpretation challenging and increases the risk of the condition going undiagnosed, in particular because Eagle syndrome is traditionally the domain of otorhinolaryngologists and underrecognized in neurological literature.

Here, our main objective was to highlight the significance of Eagle syndrome for clinical neurological practice by systematically collecting the evidence of the presentations and outcomes of Eagle syndrome.

## SURVEY METHODOLOGY

This systematic review was conducted according to standard systematic review methodology (Preferred Reporting Items for Systematic Reviews and Meta-analyses/PRISMA) and pre-registered at PROSPERO (2021 CRD42021289393).

Using the PICO approach, the following primary review question was phrased: In patients with Eagle syndrome, defined as symptoms attributed to a radiologically verified elongation of the styloid process and/or stylohyoid calcification, what are the most common neurological and non-neurological symptoms at initial presentation and what is the clinical outcome following surgical resection compared to non-surgical therapies?

We searched multiple databases for relevant literature published between January 2000 and December 2023, including PubMed, EMBASE (OVID), Web of Science (core), and the Cochrane Central Register at Controlled Trials (CENTRAL). The search terms were as follows: ("Eagle syndrome" or "Eagle's syndrome" or "Eagles syndrome" or "Eagle jugular syndrome" or "Stylo-carotid syndrome" or "styloid syndrome" or "stylohyoid syndrome" or "stylo-stylohyoid syndrome" or "styloid neuralgia" or "styloidectomy" or "styloid resection" or "elongated styloid" or "stylohyoid mineralization" or "stylohyoid calcification" or "stylohyoid ossification"). The search was supervised by a university librarian at the University of Copenhagen. We included any prospective or retrospective observational studies, including case-reports of adult human patients (>18 years) with Eagle syndrome, defined as symptoms attributed to an elongation of the stylohyoid process and/or stylohyoid ossification.

Only studies that allowed assessment of patient data on single-subject level and reported radiologically verified abnormalities were included. Furthermore, all English reports and reports in other languages, where an English abstract and a reliable full text translation is available, published in peer-reviewed journals, were incorporated. Patients were included irrespective of comorbidities, treatments, or previous diseases.

Initial selection and screening were independently conducted by MH, SA, and EWJ. Disagreements were settled by DK. Titles were reviewed initially, followed by evaluation of the abstracts with titles suggesting that a study was of relevance. Eligible studies were identified based on their full text. After relevant studies were identified, MH extracted data from the included records.

We conducted a meta-analysis on the available numerical data using SPSS version 26 for statistical analysis, including Odds Ratios and CI, Chi-squared, Fisher exact and comparison of means where appropriate; $p < 0.05$ was considered statistically significant.

## RESULTS

### Systematic literature search

The primary database searches yielded 1,940 titles in total and, of those, 876 were assessed relevant for full text review. In total, 285 studies met inclusion criteria for the final review (*Akhaddar et al., 2010*; *Al Weteid & Miloro, 2015*; *Al-Hashim et al., 2017*; *AlAbdulwahed & O'Flynn, 2021*; *Albuquerque et al., 2017*; *Aldakkan et al., 2017*; *Aldelaimi et al., 2017*; *Altun & Camci, 2016*; *Amorim et al., 2017*; *Andrade et al., 2008*; *Anuradha et al., 2020*; *Arntzen, Slowinska & Odeh, 2018*; *Atesci, Karabacakoglu & Gulmez, 2009*; *Avitia, Hamilton & Osborne, 2006*; *Aydin et al., 2018*; *Baba et al., 2017*; *Baez-Martinez et al., 2021*; *Bafaqeeh, 2000*; *Baharudin, Rohaida & Khairudin, 2012*; *Bahgat, Bahgat & Bahgat, 2012*; *Bakshi, 2016*; *Bal et al., 2018*; *Bareiss et al., 2017*; *Becker & Pfeiffer, 2013*; *Bedajit et al., 2014*; *Beder, Ozgursoy & Karatayli Ozgursoy, 2005*; *Beder et al., 2006*; *Bedi et al., 2019*; *Benet-Muñoz et al., 2017*; *Bensoussan, Letourneau-Guillon & Ayad, 2014*; *Bertossi et al., 2014*; *Betances Reinoso et al., 2015*; *Blackett et al., 2012*; *Blythe, Matthews & Connor, 2009*; *Boscainos et al., 2004*; *Bouzaidi et al., 2013*; *Boysen-Osborn & Chin, 2013*; *Brassart et al., 2020*; *Bremmer, Sergent & Ashurst, 2018*; *Buchaim, Buchaim & Shinohara, 2012*; *Budincevic, Milosevic & Pavlovic, 2018*; *Cam, Kocdor & Ozluoglu, 2020*; *Casale et al., 2008*; *Cernea et al., 2007*; *Chang et al., 2015*; *Chauhan et al., 2018*; *Chebbi et al., 2014*; *Chiang et al., 2004*; *Chiesa-Estomba, Vargas & Gonzalez-Garcia, 2021*; *Choi, 2018*; *Chourdia, 2002*; *Christopher, Gopal & Vardhan, 2020*; *Chuang et al., 2007*; *Cohn & Scharf, 2018*; *Constanzo, Ramina & Coelho Neto, 2021*; *Costantinides et al., 2016*; *Costello et al., 2010*; *Czako et al., 2019*; *Dabrowski, Ghali & Cotelingam, 2022*; *Dao et al., 2011*; *Dashti et al., 2012*; *David, Lieb & Rahimi, 2014*; *de Barros et al., 2021*; *de Souza Carvalho et al., 2009*; *Demeter et al., 2013*; *Demirtas et al., 2016*; *Devasia et al., 2004*; *Dewan et al., 2016*; *Diab et al., 2020*; *Diamond et al., 2001*; *Ding et al., 2020*; *Dong et al., 2014*; *Dou et al., 2016*; *Duarte-Celada et al., 2021*; *Dunn-Ryznyk & Kelly, 2010*; *Elimairi et al., 2015*; *Elmas & Shrestha, 2017*; *Emary, Dornink & Taylor, 2017*; *Entezami et al., 2021*; *Esiobu et al., 2018*; *Evlice et al., 2013*; *Faivre et al., 2009*; *Farina et al., 2021*; *Ferreira et al., 2014*; *Ferreira Santos & Rodrigues, 2016*; *Fini et al., 2000*; *Galletta et al., 2019a, 2019b*; *Glauser, Detchou & Choudhri, 2021*; *Goomany et al., 2020*; *Gooris et al., 2014*; *Green, Browske & Rosenthal, 2014*; *Guimarães et al., 2014*; *Gupta et al., 2021*; *Hafner, Petersson & Olsen, 2010*; *Hagiya et al., 2015*; *Hamade et al., 2016*; *Han, Kim & Yang, 2013*; *Hao et al., 2020*; *Heim et al., 2017*; *Hernandez & Velasco, 2008*; *Hernandez & Velasco, 2009*; *Ho et al., 2015*; *Hoffmann et al., 2013*; *Hopp et al., 2021*; *Horio et al., 2020*; *Hossein et al., 2010*; *Ikenouchi et al., 2020*; *Ishaq et al., 2018*; *Jain et al., 2012, 2011*; *Jelodar et al., 2018*; *Jewett & Moriarity, 2014*; *Jo et al., 2017*; *Johnson, Rosdy & Horton, 2011*; *Joshi, 2010*; *Kabak et al., 2020*; *Kadakia et al., 2018*; *Kamal et al., 2014*; *Kapoor, Jindal & Garg, 2015*; *Kar et al., 2013*; *Katsuno et al., 2014*; *Kavi & Lahiri, 2016*; *Kawasaki, Hatashima & Matsuda, 2012*; *Kay, Har-El & Lucente, 2001*; *Kesav et al., 2020*; *Keshelava, Kurdadze & Tsiklauri, 2021*; *Khandelwal, Hada & Harsh, 2011*; *Kim, Hansen & Frizzi, 2008*; *Kiralj et al., 2015*; *Kircher et al., 2013*; *Kirchhoff et al., 2006*; *Klaus et al., 2020*; *Klecha et al., 2008*; *Kouki & Guerfel, 2013*; *Kusunoki et al., 2016*; *Langlet et al., 2018*; *Langstaff, Abed & Philpott, 2012*; *Lee & Chen, 2020*; *Lee & Chung, 2020*; *Lee & Hillel, 2004*; *Lee et al., 2018*; *Lei et al., 2017*; *Lewis*

*et al., 2015; Li et al., 2018, 2019a, 2019c, 2019b; Liu, Yang & Cui, 2017; Luis Hernández, Rodríguez Sánchez & de Serdio Arias, 2014; Maamouri et al., 2022; Madaleno, Fernandes & Silva, 2016; Madden, Gross & Smith, 2015; Maggioni et al., 2009; Maher & Shankar, 2017; Maki et al., 2018; Malik, Dar & Almadani, 2015; Mann et al., 2017; Martin, Friedland & Merati, 2008; Matsumoto et al., 2012; Mattioli et al., 2021; Mayrink et al., 2012; Mejia-Vergara et al., 2022; Mendelsohn, Berke & Chhetri, 2006; Messina, 2020; Mevio et al., 2021; Michaud & Gebril, 2021; Michiels et al., 2020; Mollinedo et al., 2013; Montevecchi et al., 2019; Moon et al., 2014; More & Asrani, 2011; Morrison, Morrison & McKinstry, 2012; Mortellaro et al., 2002; Murtagh, Caracciolo & Fernandez, 2001; Mutlu & Ogul, 2017; Müderris et al., 2014; Nagato et al., 2012; Naik & Naik, 2011; Naito & Yamazaki, 2014; Nakamaru et al., 2002; Ogura et al., 2015; Ohara et al., 2012; Omami, 2019; Orlik, Griffin & Zoumberakis, 2014; Oztas & Orhan, 2012; Pace et al., 2020; Paiva et al., 2017; Pakdel et al., 2020; Papadiochos et al., 2017; Paramalingam & Kuok, 2015; Patel et al., 2020; Peng et al., 2011; Pereira et al., 2007; Permpalung et al., 2014; Peus et al., 2019; Pinheiro et al., 2013; Pithon, 2012; Pitton Rissardo & Fornari Caprara, 2019; Pokeerbux et al., 2020; Politi, Toro & Tenani, 2009; Priyamvada et al., 2021; Qureshi, Farooq & Gorelick, 2019; Radak et al., 2016; Rahman et al., 2018; Raina, Gothi & Rajan, 2009; Ramachandra, Krishnan & Reddy, 2015; Ramadan et al., 2007; Ranjan et al., 2015; Razak, Short & Hussain, 2014; Saccomanno et al., 2018; Saccomanno et al., 2021; Sadaksharam & Singh, 2012; Salamone, Falciglia & Steward, 2004; Santos Tde et al., 2014; Savranlar et al., 2005; Scavone et al., 2019; Scheller, Eckert & Scheller, 2014; Schoeff & Mukherjee, 2014; Sekido et al., 2021; Shah & Miller, 2016; Shahoon & Kianbakht, 2008; Sham et al., 2020; Sharma, Ram & Kamal, 2016; Shimizu et al., 2021; Shin et al., 2009; Shindo et al., 2019; Sigdel, Karn & Sah, 2021; Singh et al., 2015; Skjonsberg et al., 2018; Slavin, 2002; Smoot et al., 2017; Soldati et al., 2013; Song, Ahn & Cho, 2013; Soo, Chan & Wong, 2004; Sowa-Kofta, 2020; Spalthoff et al., 2016; Stafa et al., 2005; Subedi et al., 2017; Subramaniam & Kumaresan, 2012; Subramaniam & Majid, 2003; Sukegawa et al., 2017; Sultan et al., 2019; Sultan et al., 2021; Sun, Mercuri & Cook, 2006; Suzuki et al., 2020; Sveinsson, Kostulas & Herrman, 2013; Swain & Debta, 2019; Swain et al., 2017; Swain, Vidhya & Kumar, 2020; Taheri, Firouzi-Marani & Khoshbin, 2014; Tan, Crockett & Chiu, 2019; Tanti et al., 2021; Tardivo et al., 2022; Terenzi et al., 2019; Thoenissen et al., 2015; Thomas, Viswam & Xavier, 2019; Thotappa & Doni, 2012; Tijanic, Buric & Buric, 2020; Todo et al., 2012; Tomoda et al., 2018; Torikoshi et al., 2019; Uludag et al., 2013; Valerio et al., 2012; van Schaik & de Borst, 2015; Vodopivec, Klein & Prasad, 2013; Wakoh et al., 2021; Waldock, Higgins & Axon, 2020; Walli et al., 2018; Wang et al., 2022; Wani et al., 2015; Warrier et al., 2019; Weiss & Dziegielewski, 2017; Whiting & Mandel, 2016; Williams, McKearney & Revington, 2011; Wolinska et al., 2021; Wong et al., 2011; Wooton et al., 2019; Xhaxho, Vyshka & Kruja, 2021; Yamamoto et al., 2013; Yildiray et al., 2012; Yokoya et al., 2021; Yuca et al., 2010; Zammit et al., 2018; Zeckler, Betancur & Yaniv, 2012; Zhao et al., 2019*); the first study was published in 2000 (Fig. 1). All studies were single-center case studies or case series. A representative Eagle syndrome case from the authors' own institution is shown in Fig. 2.
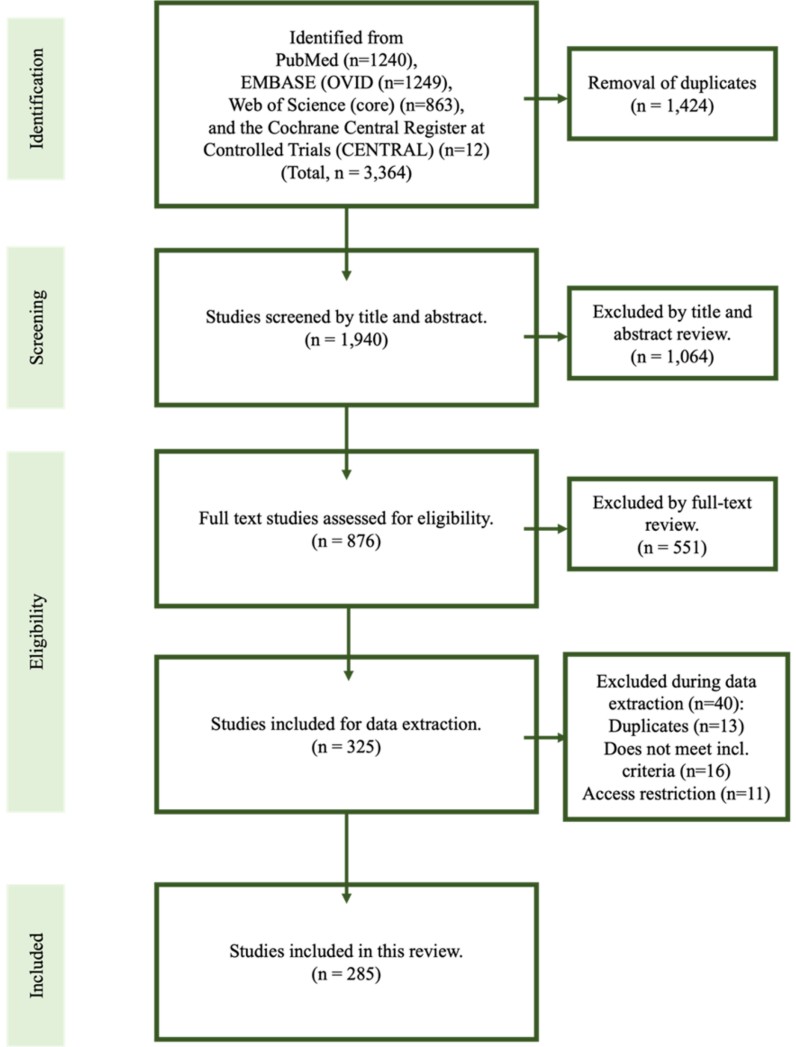

**Figure 1 Flowchart of the systematic review.** In total, 285 studies were included in this review.

## Patient population

We identified eligible data on 497 patients; mean age was 47.26 years (median 46, range 19–85). Among these, data on the sex of three patients were not available. As for the remaining cases, 246 (49.8%) were female. No correlation between age and sex was noted. The classic subvariant was most prevalent, totaling 370 patients (74.5%), compared to 117 cases (23.5%) classified as vascular. Six patients presented with both variants and for an additional four patients, the subtype was unspecified. Table 1 provides an overview of demographics and symptoms.

Pertaining to the patients with classic Eagle syndrome, no significant sex predilection was observed ($n$ = 205 women, 55.9%). Seventy-five (49.0%) of 153 classic Eagle syndrome patients for whom data were available had a history of previous cervical trauma. Tonsillectomy was the most reported ($n$ = 48, 31.4%), followed by traumatic injuries ($n$ = 20, 13.1%). With only seven (4.58%) reported cases involving abrupt cervical

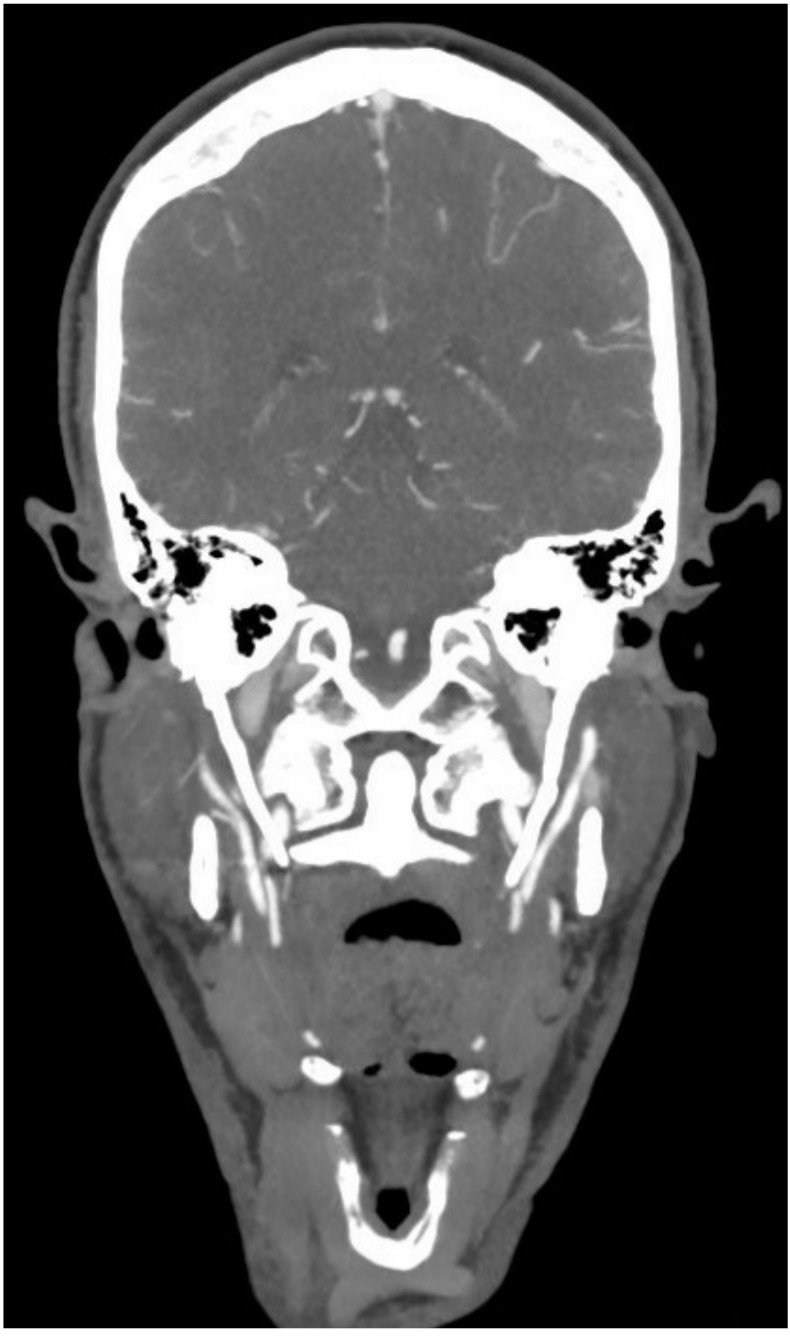

**Figure 2 Eagle syndrome.** Coronal CT angiography showing an elongated styloid process on the left (49.5 mm) and the right (42.0 mm) side. The patient was investigated for ischemic stroke (case from the authors' own institution).

movement, such as during a vigorous exercise or following a deep tissue massage, strenuous neck movements were rarely encountered in classic Eagle syndrome. Other forms of trauma described, not listed in the table due to the infrequency, include the following: radiation, parotidectomy, dental root canal treatment, tooth extraction (for each, $n = 1$), and mandibular fracture ($n = 2$).

**Table 1 Demographics and symptoms in Eagle syndrome.** Demographics and symptoms in both the total population and the classic and vascular subgroups, respectively.

| | All (incl. comb. and NS) | | Classic Eagle | | Vascular Eagle | | Significant $p$-values** |
|---|---|---|---|---|---|---|---|
| **Demographics** | | | | | | | |
| **No. (%) of patients** | 497 | (100) | 370 | (74.45) | 117 | (23.54) | <0.0001 |
| **Age, y, mean (SD)** | 47.26 | (13.95) | 46.17 | (13.42) | 51.09 | (15.02) | – |
| **Age, y, median** | 46 | | 44.5 | | 50 | | – |
| **Sex, n (%)** | (n = 494)* | | (n = 367)* | | (n = 117)* | | – |
| *Male, M* | 248 | (50.20) | 162 | (44.14) | 82 | (70.09) | |
| *Female, F* | 246 | (49.80) | 205 | (55.86) | 35 | (29.91) | |
| **Comorbidities** | (n = 72)* | (14.49) | (n = 43)* | (11.62) | (n = 29)* | (24.79) | – |
| *Cardiovascular* | 38 | (52.78) | 17 | (39.53) | 21 | (72.41) | |
| *Endocrine* | 20 | (27.78) | 11 | (25.58) | 9 | (31.03) | |
| *Cancer* | 7 | (9.72) | 7 | (16.28) | 0 | (0) | |
| *Respiratory* | 7 | (9.72) | 4 | (9.30) | 3 | (10.34) | |
| *Mental* | 6 | (8.33) | 6 | (13.95) | 0 | (0) | |
| *Neurological* | 4 | (5.56) | 2 | (4.65) | 2 | (6.90) | |
| **Previous trauma** | (n = 189)* | | (n = 153)* | | (n = 33)* | | – |
| No previous trauma | 88 | (46.56) | 74 | (48.37) | 12 | (36.36) | |
| *Tonsillectomy* | 51 | (26.98) | 48 | (31.37) | 2 | (6.06) | 0.028 |
| *Injury* | 27 | (14.29) | 20 | (13.07) | 7 | (21.21) | – |
| *Cervical movement* | 20 | (10.58) | 7 | (4.58) | 12 | (36.36) | 0.0001 |
| **Symptom debut-to-diagnosis time,** | (n = 291)* | | (n = 230)* | | (n = 61)* | | – |
| **m, mean (SD)** | 36.28 | (52.37) | 38.13 | (52.16) | 22.49 | (41.21) | 0.027 |
| **Clinical manifestations, n (%)** | (n = 497)* | | (n = 370)* | | (n = 117)* | | - |
| **Trigger/exacerbation** | | | | | | | |
| *Head rotation* | 110 | (22.13) | 88 | (23.78) | 20 | (17.09) | - |
| *Swallowing* | 63 | (12.68) | 61 | (16.49) | 2 | (1.71) | *0.002* |
| *Mandibular movements* | 56 | (11.27) | 52 | (14.05) | 3 | (2.56) | *0.005* |
| *Tongue-movement* | 7 | (1.41) | 5 | (1.35) | 2 | (1.71) | - |
| **Headache** | 83 | (16.70) | 37 | (10.00) | 43 | (36.75) | <0.0001 |
| **Cervicalgia** | 154 | (30.99) | 132 | (35.68) | 19 | (16.24) | 0.003 |
| **Speech difficulties** | 32 | (6.44) | 11 | (2.97) | 21 | (17.95) | <0.0001 |
| *Dysphonia* | 11 | (2.21) | 9 | (2.43) | 2 | (1.68) | – |
| *Dysarthria* | 11 | (2.21) | 0 | (0) | 11 | (9.40) | – |
| *Other* | 10 | (2.01) | 2 | (0.54) | 8 | (6.84) | – |
| **Dysphagia** | 115 | (23.14) | 106 | (28.65) | 7 | (5.98) | 0.0001 |
| **Odynophagia** | 58 | (11.67) | 54 | (14.59) | 2 | (1.71) | 0.003 |
| **Otalgia/hearing impairments** | 85 | (17.10) | 77 | (20.81) | 7 | (5.98) | 0.002 |
| **Tinnitus** | 25 | (5.03) | 16 | (4.32) | 8 | (6.84) | – |
| **Horner syndrome** | 16 | (3.22) | 5 | (1.35) | 11 | (9.40) | 0.0004 |
| **Sensory symptoms** | 30 | (6.04) | 8 | (2.16) | 21 | (17.95) | <0.0001 |
| **Dysgeusia** | 3 | (0.60) | 2 | (0.54) | 1 | (0.85) | – |
| **Ataxia** | 10 | (2.01) | 1 | (0.27) | 9 | (7.69) | 0.002 |

| Table 1 (continued) | | | | | | | |
| --- | --- | --- | --- | --- | --- | --- | --- |
| | All (incl. comb. and NS) | | Classic Eagle | | Vascular Eagle | | Significant *p*-values** |
| Hemiparesis/hemiplegia | 31 | (7.44) | 4 | (1.08) | 27 | (23.08) | <0.0001 |
| Aphasia | 17 | (3.42) | 0 | (0) | 17 | (14.53) | <0.0001 |
| Visual impairments | 35 | (7.04) | 4 | (1.08) | 30 | (25.64) | <0.0001 |
| Weakness | 26 | (5.23) | 2 | (0.54) | 24 | (20.51) | <0.0001 |
| (Near) syncopation | 20 | (4.02) | 3 | (0.81) | 13 | (11.11) | 0.0001 |
| Dizziness | 15 | (3.02) | 6 | (1.62) | 8 | (6.84) | – |
| Vertigo | 11 | (2.21) | 5 | (1.35) | 6 | (5.13) | – |
| Transient ischemic attack/TIA | 8 | (1.61) | 0 | (0) | 8 | (6.84) | 0.006 |
| Stroke | 16 | (3.22) | 0 | (0) | 16 | (13.68) | 0.0001 |
| Carotid dissection | 47 | (9.46) | 0 | (0) | 47 | (40.17) | 0.0001 |
| ICA (pseudo) aneurysm | 18 | (3.62) | 0 | (0) | 17 | (14.53) | 0.001 |
| ICA tortuosity | 9 | (1.81) | 0 | (0) | 9 | (7.69) | 0.005 |
| Jugular vein compression/stenosis | 27 | (5.43) | 0 | (0) | 26 | (22.22) | 0.0003 |
| Cerebral infarct | 26 | (5.23) | 0 | (0) | 26 | (22.22) | 0.0003 |
| Pharyngeal foreign body sensation | 136 | (27.36) | 128 | (34.60) | 5 | (4.27) | <0.0001 |
| Pharyngeal pain | 148 | (29.78) | 144 | (38.92) | 3 | (2.56) | <0.0001 |
| Temporomandibular/orofacial pain | 65 | (13.08) | 64 | (17.30) | 1 | (0.85) | 0.003 |
| Swelling (submandibular or cervical region) | 20 | (4.02) | 17 | (4.60) | 3 | (2.56) | – |
| Insomnia/sleep disturbance | 6 | (1.21) | 2 | (0.54) | 3 | (2.56) | – |
| Weight loss | 4 | (0.80) | 1 | (0.27) | 3 | (2.56) | – |

Notes:
* N = cases with data available.
** See Methods for statistical tests.
Comb. = Combination of both classical and vascular Eagle syndrome. NS = Not specified. ICA = Internal carotid artery.

In contrast, a male preponderance was observed in the vascular subtype ($n = 82$, 70.1%). In terms of potential prior trauma, only two vascular Eagle patients (6.1%) had undergone a tonsillectomy, seven (21.2%) had experienced a traumatic injury and, most commonly, twelve cases (36.4%) reported abrupt neck movement. Hence, cervical movements as trigger factors were more prevalent in vascular (12/33 patients) than in classic (7/153 patients) Eagle syndrome (Odds Ratio 7.95, 95% CI [2.9–21.7]; $p = 0.0001$). By contrast, a history of tonsillectomy was more frequent in classic (48/153 patients) than in vascular (2/33 patients) Eagle syndrome (Odds Ratio 5.2, 95% CI [1.2–22.4]; $p = 0.028$).

## Clinical symptomatology

There was a large variation in the time from onset of symptoms to definite diagnosis with a mean of 3 years (median 18 months, range 0–324 months). Among the 61 patients with available data in the vascular subtype, a mean diagnostic delay of 22.5 months (median 2 months) was observed. By contrast, the classic subgroup, with a total of 230 cases, resulted in a longer diagnostic delay (mean 38.1 months, median 24 months; difference 16 months, 95% CI [−30.1 to −1.8]; $p = 0.027$).

### Trigger factors

Patients also reported triggers or worsening of symptoms due to specific movement. Consistent across both classic and vascular Eagle syndrome, head rotation was frequently observed ($n$ = 88 patients, 23.8% and $n$ = 20, 17.1%, respectively), while the occurrence of exacerbation with tongue movement was less common ($n$ = 5, 1.4%, observed in the classic and $n$ = 2, 1.7%, in the vascular subgroup). Exacerbations with swallowing and mandibular movements were more frequent in patients exhibiting classic Eagle syndrome ($n$ = 61, 16.5%, and $n$ = 52, 14.1%, respectively). Overall, headache and Horner syndrome were more frequent in vascular Eagle syndrome, while dysphagia and neck pain were more prominent in classic Eagle syndrome (all $p$ < 0.01, Table 1).

### Symptoms of classic Eagle syndrome

Patients of the classic Eagle syndrome group presented with a range of symptoms and signs, including, in descending order, pharyngeal pain ($n$ = 144, 38.9%), cervical pain ($n$ = 132, 35.7%), pharyngeal foreign body sensation ($n$ = 128, 34.6%), dysphagia ($n$ = 106, 28.7%), otalgia or hearing impairments ($n$ = 77, 20.8%), temporomandibular or orofacial pain ($n$ = 64, 17.3%), odynophagia ($n$ = 54, 14.6%), headache ($n$ = 37, 10.0%), swelling ($n$ = 17, 4.6%), and tinnitus ($n$ = 16, 4.3%). Furthermore, 11 cases (3.0%) reported speech difficulties, hereof 9 (2.4%) with dysphonia. A further eight patients (2.16%) experienced sensory symptoms.

### Symptoms of vascular Eagle syndrome

Symptoms and signs in the vascular subgroup included, in decreasing frequency, headache ($n$ = 43, 36.8%), visual impairments ($n$ = 30, 25.6%), hemiplegic or -paretic conditions ($n$ = 27, 23.1%), weakness ($n$ = 24, 20.5%), and speech difficulties ($n$ = 21, 18.0%), hereof 11 (9.4%) with dysarthria, only 2 (1.7%) with dysphonia, and an additional 8 (6.8%) with other types of speech impediments, such as slurred speech. Moreover, the following clinical manifestations were also observed: sensory symptoms ($n$ = 21, 17.8%), cervical pain ($n$ = 19, 16.24%), aphasia ($n$ = 17, 14.5%) (near)syncope ($n$ = 13, 11.1%), Horner's syndrome ($n$ = 11, 9.4%), ataxia ($n$ = 9, 7.7%), tinnitus ($n$ = 8, 6.8%), dizziness ($n$ = 8, 6.8%) and vertigo ($n$ = 6, 5.1%). Pharyngeal pain, pharyngeal foreign body sensation, temporomandibular or orofacial pain, odynophagia and swelling occurred with $n \leq 5$.

Forty-seven vascular Eagle syndrome patients (40.2%) had a documented carotid dissection, 17 (14.5%) had an internal carotid artery aneurysm and internal carotid artery tortuosity was documented in nine cases (7.7%). A total of 26 patients (22.2%) had jugular vein involvement, including compression and stenosis.

### Investigations

Vascular patients were predominantly investigated using imaging modalities like CT angiography ($n$ = 72, 62.6%) and plain CT ($n$ = 37, 32.2%), while both CT ($n$ = 273, 74.2%) and orthopantomogram ($n$ = 119, 32.3%) were frequently utilized for classic Eagle syndrome patients. Bilaterally elongated styloid processes were noted in 253 patients (55.5%). A right-sided unilateral elongated styloid process was reported in 76 patients

**Table 2 Diagnosis of Eagle syndrome.** Diagnosis in both the total population and the classic and vascular subgroups, respectively.

| | All (incl. comb. and NS) | | Classic Eagle | | Vascular Eagle | | Significant p-values** |
|---|---|---|---|---|---|---|---|
| **Diagnosis, n (%)** | | | | | | | |
| **Imaging modality** | (n = 493)* | | (n = 368) | | (n = 115)* | | |
| Computed Tomography (CT) | 318 | (64.50) | 273 | (74.18) | 37 | (32.17) | <0.0001 |
| C. angiography | 75 | (15.21) | 3 | (0.82) | 72 | (62.61) | <0.0001 |
| Multidetector CT angiography | 3 | (0.61) | 2 | (0.54) | 1 | (0.87) | – |
| Orthopantomography | 122 | (24.75) | 119 | (32.34) | 2 | (1.74) | <0.0001 |
| Magnetic Resonance Imaging (MRI) | 5 | (1.01) | 2 | (0.54) | 3 | (2.61) | – |
| Conventional angiography | 1 | (0.20) | 1 | (0.27) | 0 | 0 | – |
| X-ray | 24 | (4.87) | 22 | (5.98) | 1 | (0.87) | <0.0001 |
| **Objective findings** | (n = 399)* | | (n = 287)* | | (n = 106)* | | |
| Isolated elongated styloid process | 262 | (65.66) | 181 | (63.07) | 77 | (72.64) | – |
| Concomitant stylohyoid Mineralization | 137 | (34.34) | 106 | (36.93) | 29 | (27.36) | – |
| **Side** | (n = 456)* | | (n = 347)* | | (n = 101)* | | |
| Bilateral | 253 | (55.48) | 185 | (64.46) | 64 | (63.37) | – |
| Unilateral/Right | 76 | (16.67) | 56 | (16.14) | 19 | (18.81) | – |
| Unilateral/Left | 81 | (17.76) | 61 | (17.58) | 17 | (16.83) | – |
| Unilateral/NS | 46 | (10.09) | 45 | (12.97) | 1 | (0.99) | <0.001 |
| **Styloid process length, mm, mean (SD)** | | | | | | | |
| Right | 45.7 | (24.45) | 46.05 | (24.43) | 45.33 | (24.98) | – |
| Left | 45.63 | (23.94) | 45.48 | (23.82) | 45.68 | (23.93) | – |

Notes:
* N = cases with data available.
** see Methods for statistical tests.
Comb. = combination of both classical and vascular Eagle syndrome. NS = not specified.

(16.7%), a left-sided one in 81 patients (17.8%), and this was not specified in an additional 46 patients (10.1%). The length of right styloid processes (mean 45.7 mm, 206 cases) was identical to that of left processes (mean 45.6 mm, 182 cases). Furthermore, 181 cases (63.1%) had an isolated elongated styloid process, and 106 cases (36.9%) had concomitant stylohyoid mineralization. The same figures for the vascular subgroup were 77 (72.6%) and 29 (27.4%), respectively. Table 2 offers an overview of the diagnostic investigations and their relevant findings.

## Treatments

The treatment options include both medical and surgical interventions. Table 3 provides an overview of treatment, outcomes, and complications in both the total population and the classic and vascular subgroup, respectively. The most frequent intervention across subgroups was styloidectomy; a surgical procedure encompassing either partial or complete styloid process-complex resection. In the classic subgroup, 267 patients (72.2%) underwent this treatment modality and, of those, 158 (59.2%) were treated with a transoral approach, 87 (32.6%) with a transcervical approach, and three patients (1.1%) underwent both. One-fourth of the patients (n = 92, 24.9%) were medically managed, with the most prevalent medications being: non-steroidal anti-inflammatory drugs (n = 39, 42.4%),

**Table 3 Treatment and outcomes in Eagle syndrome.** Treatment, outcomes, and complications in both the total population and the classic and vascular subgroups, respectively.

| | All (incl. comb. and NS) | | Classic Eagle | | Vascular Eagle | | Significant *p*-values** |
|---|---|---|---|---|---|---|---|
| **Treatment and outcome, n (%)** | | | | | | | |
| **None/NS** | 44 | (8.85) | 37 | (10.0) | 7 | (5.98) | – |
| **Medical** | (*n* = 153)* | (30.78) | (*n* = 92)* | (24.86) | (*n* = 57)* | (48.72) | |
| *NSAID* | 58 | (37.91) | 39 | (42.39) | 12 | (21.05) | – |
| *Antiplatelet/anticoagulant* | 42 | (27.45) | 0 | (0) | 40 | (70.18) | <0.0001 |
| *Anticonvulsants* | 28 | (18.30) | 27 | (29.35) | 0 | (0) | <0.0001 |
| *  - Carbamezapine* | 17 | (11.11) | 17 | (18.48) | 0 | (0) | <0.0001 |
| *  - Gabapentin* | 9 | (5.88) | 9 | (9.78) | 0 | (0) | <0.05 |
| *  - Pregabalin* | 3 | (1.96) | 3 | (3.26) | 0 | (0) | – |
| *  - Valproate acid* | 3 | (1.96) | 2 | (2.17) | 0 | (0) | – |
| *Corticosteroids* | 12 | (7.84) | 11 | (11.96) | 1 | (0.02) | <0.05 |
| *Psychotropics* | 11 | (7.19) | 10 | (10.87) | 0 | (0) | – |
| *Analgesics* | 43 | (28.10) | 36 | (39.13) | 4 | (7.02) | <0.001 |
| *Muscle relaxant* | 13 | (8.50) | 13 | (14.13) | 0 | (0) | <0.01 |
| *Outcome* | (*n* = 123)* | | (*n* = 74)* | | (*n* = 46)* | | |
| *  - No improvement* | 42 | (34.15) | 28 | (37.84) | 12 | (26.09) | – |
| *  - Improvement* | 54 | (43.90) | 29 | (39.19) | 25 | (54.35) | – |
| *  - Symptom resolution* | 27 | (21.95) | 17 | (22.97) | 9 | (19.57) | – |
| **Surgical** | (*n* = 341)* | (68.61) | (*n* = 267)* | (72.16) | (*n* = 65)* | (55.56) | |
| *Transoral* | 161 | (47.21) | 158 | (59.18) | 1 | (1.54) | <0.0001 |
| *Transcervical* | 138 | (40.47) | 87 | (32.58) | 45 | (69.23) | <0.001 |
| *Both* | 3 | (0.88) | 3 | (1.12) | 0 | (0) | – |
| *NS* | 39 | (11.44) | 19 | (7.12) | 19 | (29.23) | – |
| *Right* | 75 | (21.99) | 51 | (19.10) | 23 | (35.38) | – |
| *Left* | 81 | (23.75) | 60 | (22.47) | 18 | (27.69) | – |
| *Bilateral* | 97 | (28.45) | 78 | (29.21) | 17 | (26.15) | – |
| *NS* | 88 | (25.81) | 78 | (29.21) | 7 | (10.77) | <0.05 |
| *Styloid process length after, R, mm, mean* | 19.91 | (4.54) | 20.52 | (4.84) | 17.8 | (3.64) | – |
| *Styloid process length after, L, mm, mean* | 24.06 | (5.54) | 23.02 | (5.70) | 34.5 | (5.21) | – |
| *Complications* | 26 | (7.62) | 15 | (5.62) | 10 | (15.38) | – |
| *Outcome* | (*n* = 320)* | | (*n* = 254)* | | (*n* = 58)* | | |
| *  - No improvement* | 7 | (2.19) | 6 | (2.36) | 1 | (1.72) | – |
| *  - Improvement* | 91 | (28.44) | 57 | (22.44) | 31 | (53.45) | <0.001 |
| *  - Symptom resolution* | 222 | (69.38) | 191 | (75.20) | 26 | (44.83) | <0.05 |
| **Comb. of both medical and surgical** | (*n* = 60)* | (12.07) | (*n* = 36)* | (9.73) | (*n* = 21)* | (17.95) | |
| *Based on the data above* | | | | | | | |
| Comb. treatment w. overall improvement | 17 | (28.33) | 7 | (19.44) | 9 | (42.86) | – |
| No effect medical, successful surgical | 26 | (43.33) | 21 | (58.33) | 5 | (23.81) | – |
| Comb. treatment w. NS overall outcome | 17 | (28.33) | 8 | (22.22) | 7 | (33.33) | – |

| Table 3 (continued) | | | | | | | |
| --- | --- | --- | --- | --- | --- | --- | --- |
| | All (incl. comb. and NS) | | Classic Eagle | | Vascular Eagle | | Significant *p*-values** |
| **Treatment and outcome, n (%)** | | | | | | | |
| **Other (indirect) treatments** | | *Improvement*: | | *Improvement*: | | *Improvement*: | |
| *ICA or jugular vein stenting* | 23 | 22 | 0 | 0 | 23 | 22 | - |
| *Thrombectomy* | 9 | 7 | 0 | 0 | 9 | 7 | - |
| *Carotid endarterectomy* | 3 | 3 | 0 | 0 | 3 | 3 | - |
| *Physical therapy* | 14 | 11 | 11 | 8 | 3 | 3 | - |
| *Cervicotomy* | 13 | 12 | 2 | 2 | 11 | 10 | - |

Notes:
  * N = cases with data available.
  ** See Methods for statistical tests.
  Comb. = combination of both classical and vascular Eagle syndrome. ICA = internal carotid artery. NS = not specified. NSAID = non-steroidal anti-inflammatory drug.

analgesics (*n* = 36, 39.13%) and anticonvulsant (*n* = 27, 29.4%). Following the latter, carbamazepine (*n* = 17) and gabapentin (*n* = 9) were the most widely used. An additional 11 patients tried physical therapy.

In the vascular subgroup, a significantly larger subset (*n* = 57, 48.7%) received medical treatment, including 40 individuals (70.2%) on antiplatelet/-coagulant medication and an additional 12 (21.1%) on non-steroidal anti-inflammatory drugs. Of the 65 surgical cases, only one patient (1.5%) underwent a transoral approach and 45 patients (69.2%) underwent a transcervical approach. The approach of the remaining 19 cases (29.3%) was not specified. Other treatment options for patients with vascular symptoms include internal carotid artery/jugular vein stenting (*n* = 23), thrombectomy (*n* = 9) and carotid endarterectomy (*n* = 3).

## Outcomes

Outcome data were available for 434 patients. In most instances, the outcome was measured as subjective well-being and symptom resolution at the last available follow-up post-treatment. In a minority of cases, it was instead measured as restoration of vascular flow. Surgically treated patients achieved overall better outcomes than medically treated ones: Eighty-one (65.9%) of 123 medically treated patients experienced improvement or complete resolution, while the same applied to 313 (97.8%) of 320 surgical patients (Odds Ratio 1.49; 95% CI [1.08–2.05]).

Data on the outcome of medical treatment for patients in the classic subgroup were available for 74 patients. Forty-six cases (62.2%) experienced either improvement or complete resolution of their symptoms. Among the 28 (37.8%) patients, whose symptoms did not respond to medical treatment, 21 had subsequently successful surgical treatment.

Data in the outcome of medical treatment for patients in the vascular subgroup were available for 46 patients. Thirty-four cases (73.9%) experienced either improvement or complete resolution of their symptoms, while no effect was seen in 12 (26.1%). Of the latter group, five were further treated surgically with successful outcome.

The overall outcome of surgical intervention in the total population exhibited a high success rate with improvement in 91 cases (28.4%) and complete resolution in 222 cases (69.4%). Of the seven cases (2.2%) with lack of effect, five had been treated with the transcervical, one with the transoral and one with a combined approach.

In cases involving vascular issues, secondary manifestations of Eagle syndrome, such as carotid dissections, ischemic stroke, and aneurysms, were common and required treatment different than the abovementioned approaches. These interventions, both performed independently and as a part of a comprehensive treatment plan, yielded overall good results: Twenty-two of the 23 patients who were treated with internal carotid artery/jugular vein stenting experienced improvement of symptoms as did seven of the nine patients who underwent thrombectomy.

Sequelae after styloidectomy was described in a total of 26 patients (7.6%), including the following: facial nerve weakness, bleeding, paresis of mouth/lips, auricular paresthesia, and hypoglossal neuropraxia, the majority of which were minor or no longer present at last follow-up.

## DISCUSSION

Although case occurrences of Eagle syndrome have been frequently reported in the literature, studies regarding overall patient demographics, symptoms, treatments, and outcomes are still lacking. Based on our data from a total population of 497 cases, the average patient with Eagle syndrome is in their fourth or fifth decade of life at the time of diagnosis. With a rough ratio of 3:1, the classical Eagle syndrome is more prevalent than the vascular variant. Based on the obtained data, clear distinctions in the clinical picture of the two phenotypes have been identified. The symptomatic presentation of the vascular type is more likely to be characterized by headache, neurovascular events, and a shorter diagnostic delay. The natural history data from the present study also suggests a more acute progression of the disease. In contrast, patients with the classical variant are more prone to experience symptoms of nonspecific nature such as neck pain, globus sensation and dysphagia. With a high success rate, surgical management with styloidectomy is often considered the definitive treatment of Eagle syndrome.

### Classic *vs.* vascular Eagle syndrome

The two forms of Eagle syndrome were compared for differences in demographics. The average age was 47.26 ± 13.95 years and no significant deviation was noted across the two subgroups, but the male-female ratio varied. Given an approximate ratio of 1:1, classic Eagle syndrome may not show sex-specific tendencies, while the vascular ratio of 2:1 could potentially suggest an association.

A second difference between the two pertains to the potential trigger of the initial styloid process remodeling. Though many theories have been put forth regarding the pathogenesis, there appears to be consensus regarding a correlation between a prior iatrogenic pharyngeal procedure, like a tonsillectomy, and a subsequent development of Eagle syndrome in some patients (*Wong et al., 2011*). This correlation may be explained by a hypothesis developed by Steinmann (*Saccomanno et al., 2021*). Herein, it is suggested

that a traumatic event can trigger two mechanisms (either reactive hyperplasia or reactive metaplasia) in an initially nonconforming styloid process, which may then lead to lengthening of the process and the ossification of the stylohyoid ligament. (Another proposition is the elongated styloid process as an anatomical variant which may present a risk factor in case of trauma.) A history of tonsillectomy was present in one-third of the classic cases and is, thus, consistent with the existing literature. The same, however, did not apply to the patients belonging to the vascular group with only six percent tonsillectomized individuals. By contrast, abrupt cervical movement, neck manipulation and overextension as trigger factors were more prevalent here. In a single center-case control study conducted by Tardivo, it was concluded that the styloid process angulation on the coronal plane tends to be more acute in patients with a styloid process related carotid dissection (*Tardivo et al., 2022*). Furthermore, it was found that the styloid-C1 distance is significantly shorter at the side of dissection. These findings might explain why individuals with vascular Eagle syndrome exhibit increased sensitivity to strenuous and vigorous cervical movements as multiple factors, beyond just the physical elongation of the process, seem to contribute to the arterial impingement.

The medical history is a significant factor in diagnosing Eagle syndrome. Based on the obtained data, it is evident that the two phenotypes present with different distinct clinical features. Patients with classical Eagle will often wait longer before seeking medical attention. The clinician then encounters symptoms of more diffuse nature, including pain located to the post auricular region, temporomandibular region, cervical region along with pharyngeal globus sensation and dysphagia. These symptomatic manifestations are described in the existing literature and have been attributed to the following pathophysiological mechanisms: compression of neural elements or irritation of the pharyngeal mucosa by the styloid process, ossified stylohyoid ligament or degenerative and inflammatory changes in the tendonous portion of the stylohyoid insertion (*Khandelwal, Hada & Harsh, 2011*).

Not surprisingly, Eagle syndrome in its vascular form has a notably shorter diagnostic delay (by approximately 16 months compared to the classic group). The clinical presentation is characterized by neurovascular symptoms and signs such as focal neurological deficits, attributable to the impingement of the adjacent vessels producing irritation of the sympathetic nerves in the arterial sheath or compromising the vascular circulation, which may require more urgent medical care and attention (*Khandelwal, Hada & Harsh, 2011*).

## Treatment options for Eagle syndrome

Diagnosis is confirmed by radiological investigation. Plain CT and CT angiography are considered the golden standards in terms of diagnostic workup as they allow precise visualization and analysis of the anatomical structures.

The management of Eagle syndrome often involves a dichotomy between conservative methods of pharmacotherapy and a more definitive surgical option. Findings here suggest that the medical treatment of classic Eagle syndrome is far more pain-centered, including NSAIDs, anticonvulsant, psychotropics and general analgesics, while the treatment of

patients with vascular Eagle syndrome focuses, to a greater extent, on avoiding potential vascular compromise, including antiplatelet/-coagulant therapies. Outcome of these treatments were moderately satisfactory. When medical treatment yields unsatisfactory results, patients should be referred for surgical management.

The surgical options consist of two approaches: the transoral and the transcervical routes. Advantages for the former include fewer postoperative cosmetic deformities and the relative ease of performance, though visualization may be poorer and the risk of causing damage to the adjacent neurovascular structures is increased (*Sukegawa et al., 2017*). Drawing from the findings here, this treatment approach is favored for classical cases, while hardly any vascular patients were treated in this manner; perhaps due to the pre-existing vulnerable state of the neurovascular structures. Instead, vascular patients are more likely to undergo a transcervical approach which may have a longer operative duration and postoperative cosmetic issues but in return provides an adequate field of operation with good surgical access to the styloid process and its surrounding tissues, hence avoiding unnecessary harm to adjacent structures (*Sukegawa et al., 2017*).
The results showed a notably low occurrence of adverse events associated with the surgical intervention. These outcomes suggest a relatively favorable safety profile. However, transpharyngeal manipulation with a manual fracture of the elongated styloid process does not usually relieve symptoms and risks damage to adjacent neurovascular structures. The diversity of treatment options for Eagle syndrome reflects the absence of a standardized protocol. The lack of uniform guidelines may lead clinicians to rely on a broad range of methods and medical approaches based on individual observations and abilities.

## Limitations and strengths

Systematic reviews are prone to certain limitations and the findings should be interpreted in the light of these. For instance, the inclusion of a patient cohort, where incomplete data for the majority of the patients is inevitable, may limit the depth of analysis and thereby, potentially affect the overall outcomes. The incorporation of single cases or small case series, often without a clear statement about potentially excluded patients and the reasons for their exclusions, may introduce biases. These include selection bias associated with factors like age and comorbidities and publication bias where cases that are successfully managed are more likely to be reported. Together with the retrospective nature of the data, this could all impact the generalizability of our conclusions.

Despite these limitations, this review has important strengths. These include the size of the cohort and the systematic methodology (PRISMA) along with preregistration at PROSPERO, which contribute to the review's credibility and reliability. Since a comprehensive and up-to-date review on Eagle syndrome has not been done before, the article unveils novel perspectives. For example, Eagle syndrome is being reported to have a female preponderance, and while we observed this to some extent in the classic subgroup, there was a significantly higher prevalence of males in the vascular subgroup (*Bareiss et al., 2017*). We hope the detailed information of the diverse neurological presentations revealed

here may be of clinical value for neurologists and raise the awareness for good outcome with surgical treatment.

## CONCLUSIONS

In this systematic review, we have collected data on the correlation between classic and vascular Eagle syndrome with demographics, symptoms, treatments, and outcomes. Based on the available data, clear distinctions between the two types of Eagle syndrome have been identified. Due to the underlying pathogenetic mechanism and the different anatomical structures affected by the styloid process, the clinical presentations vary greatly. On the one hand, a male preponderance was observed in the vascular Eagle syndrome and cervical movements as trigger factors were prevalent. Furthermore, headache, Horner syndrome and focal neurological deficits were more frequent. This, along with the notably shorter diagnostic delay, convincingly show that vascular Eagle syndrome is a condition with potentially serious neurovascular complications. On the other hand, symptoms such as dysphagia and neck pain were more prominent in classic Eagle syndrome, as was a history of tonsillectomy. Overall, surgical treatment achieves better outcomes than medical therapy. Surgical styloidectomy demonstrates overall promising effect and a relatively favorable safety profile, so it might be considered the definitive treatment of Eagle syndrome. The implementation of a standardized protocol for managing Eagle syndrome would benefit both patients and healthcare providers. Although traditionally the domain of the otorhinolaryngologist, neurologists should remember Eagle syndrome for its varied neurological presentations and effective treatment options.

### Funding
The authors received no funding for this work.

### Competing Interests
The authors declare that they have no competing interests.

### Author Contributions
- Melika Hassani performed the experiments, analyzed the data, prepared figures and/or tables, authored or reviewed drafts of the article, and approved the final draft.
- Elisabeth Waldemar Grønlund performed the experiments, authored or reviewed drafts of the article, and approved the final draft.
- Simon Sander Albrechtsen performed the experiments, authored or reviewed drafts of the article, and approved the final draft.
- Daniel Kondziella conceived and designed the experiments, analyzed the data, prepared figures and/or tables, authored or reviewed drafts of the article, and approved the final draft.

### Data Availability
This is a systematic review/meta-analysis.

## Supplemental Information

Supplemental information for this article can be found online at http://dx.doi.org/10.7717/peerj.17423#supplemental-information.

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
