# Peer review of "Neurological phenotypes and treatment outcomes in Eagle syndrome: systematic review and meta-analysis"

_PeerJ, doi:10.7717/peerj.17423_

## Round 0.1 · original submission · Minor Revisions

Thank you for submitting your review on the neurological manifestations and treatment options of Eagle syndrome. It addresses a rare but important symptom and the methodological quality of the manuscript is high.

There are only two minor comments in addition to the reviewers' comments:
The abbreviations in Table 2 should be explained.
The source of Figure 2 should be indicated.

·

Basic reporting

The authors aimed to systematically analyze the available scientific evidence of the clinical presentations and outcomes of Eagle syndrome, an underrecognized condition caused by an elongated styloid process and/or calcification of the stylohyoid ligament compressing the adjacent vascular and neuronal structures. The methodology is described in detail, is based on a PICO question, and follows the PRISMA guidelines.

Experimental design

The authors identified 497 cases, 75% had the classic variant. The results are presented straight forward and discussed appropriately.

Validity of the findings

This is a very important contribution and I only have minor issues to be considered:
- Lane 265: rephrase the term “cerebral deficits”
- Lanes 265/266: the source of “natural history data” is not stated – current study or other?
- Results: I wonder whether there has been a selection bias for either treatment approach, e.g. age, comorbidities etc.
- Limitations: discuss a potential bias for reporting of more severe cases and ones with aggressive treatment approaches

Additional comments

None

Reviewer 2 ·

Basic reporting

No comment

Experimental design

No comment

Validity of the findings

P value should be added for each item in the tables.

Reviewer 3 ·

Basic reporting

In the text spanning lines 59 and 60, it was initially stated that Watt Eagle in 1937 first identified the link between pain and an elongated styloid process. However, it has been clarified that Dr. Eagle was not the pioneer in recognizing this correlation. According to the recent article titled "Minimally Invasive Cervical Styloidectomy in Stylohyoid Syndrome (Eagle Syndrome) - 10.3390/jcm12216763," it is documented that Dwight, as early as 1907, made a seminal observation about the association between orofacial pain and variations in the stylohyoid ligament's anatomy, offering one of the initial insights into the pathogenesis of the condition. Furthermore, in 1927, Garel conducted independent symptom analysis, and Bernfeld, in 1932, introduced the condition's first successful surgical intervention. The syndrome, occasionally referred to in literature as the Garel–Bernfeld syndrome, reflects the contributions of these earlier researchers to the understanding and treatment of the disease, predating Eagle's description.

It should be noted that predicting the overall prevalence is nearly impossible due to the significant variation in diagnostic criteria across publications. The figures cited in the article, including a 4% prevalence of elongated styloids in the population and 4% exhibiting symptoms, were presented by Dr. Eagle but have been challenged by the scientific community. To address the most recent findings, not including case reports. As an example one can refer to the article titled "Elongated Styloid Process in Panoramic Radiographs" published in the Journal of Health Sciences in 2021.

Experimental design

no comment

Validity of the findings

no comment

Additional comments

The article is indeed intriguing, featuring a clear structure and well-defined premises. It encompasses a wide range of publications and serves as a comprehensive summary of the current knowledge regarding Eagle's syndrome. This approach enriches the understanding of the syndrome's diagnostic and treatment strategies and highlights the evolution of scientific thought on the topic. By systematically reviewing extensive literature, the article effectively consolidates existing research, offering a valuable resource for both clinicians and researchers interested in Eagle's syndrome. What is most important for clinicians it importantly highlights that styloidectomy as a surgical management approach, known for its high success rate, is deemed the definitive treatment for Eagle syndrome. Crucially for clinicians, the article underscores the significance of styloidectomy. This surgical intervention, noted for its elevated success rates, is recognized as the quintessential remedy for Eagle syndrome.

---

## Round 0.2 · accepted · Accept

The authors have adequately addressed all the comments of the reviewers and the editor. I congratulate the authors on this well-written, interesting and relevant manuscript.

·

Basic reporting

no comment

Experimental design

no comment

Validity of the findings

no comment

Additional comments

Congratulations for performing this excellent study!